# Evolutionary plasticity and functional repurposing of the essential metabolic enzyme MoeA
Daniela Megrian [1,2], Mariano Martinez[1,3], Pedro M. Alzari [1] & Anne Marie Wehenkel [1,3]

MoeA, also known as gephyrin in higher eukaryotes, is an enzyme essential for molybdenum cofactor (Moco) biosynthesis and involved in GABA and GlyR receptor clustering at the synapse in animals. We recently discovered that Actinobacteria have a repurposed version of MoeA (Glp) linked to bacterial cell division. Since MoeA exists in all domains of life, our study explores how it gained multifunctionality over time. We use phylogenetic inference and protein structure analyses to study its diversity and evolutionary history. Glp-expressing Bacteria have at least two copies of the gene, and analysis of their putative active sites suggests that Glp lost its enzymatic role. In Archaea, we find an ancestral duplication, with one paralog that may bind tungsten instead of molybdenum. Early eukaryotes acquired MoeA from Bacteria, MogA fused with MoeA in the opisthokont ancestors, and it finally gained roles in anchoring inhibitory neurotransmitters. Our findings highlight MoeA's functional versatility and repurposing.

Most biological processes in all domains of life require redox reactions that rely on enzymes with associated metal ions or bound metal cofactors[1]. Available evidence suggests that the last universal common ancestor (LUCA) used these cofactors, including the pterin-based molybdenum cofactor (Moco)[2]. Moco is an essential component of a group of redox enzymes known as molybdoenzymes. Moco is essential for most living organisms, as the versatile redox chemistry of molybdenum allows molybdoenzymes to catalyze important reactions in biochemical cycles of carbon, nitrogen, and sulfur[3]. Many pathogenic bacteria such as *Mycobacterium tuberculosis*, *Salmonella enterica*, *Campylobacter jejuni* or *Haemophilus influenzae* use molybdoenzymes (sulfite dehydrogenases, S/N-oxides reductases, nitrate reductases, formate dehydrogenases) to facilitate adaptation of the pathogen to its environment by supporting energy generation, or by converting compounds generated in the host during inflammation[4]. In humans, deficiency in Moco biosynthesis causes a rare disease responsible for the loss of the enzymatic activity of all molybdoenzymes, leading to severe neurological damage and premature death[5].

Moco biosynthesis comprises three steps catalyzed by enzymes that are well-conserved in all domains of life[6]. Interestingly, secondary functions have been identified for proteins involved in Moco biosynthesis in animals[7] and more recently in *Corynebacteriales*[8]. Gephyrin is a well-documented moonlighting protein that was first discovered for its role in post-synaptic signaling[9] and only later identified as a fusion protein composed of MoeA

and MogA, the two enzymes responsible for the last step in Moco biosynthesis[10]. This gene fusion allowed gephyrin to acquire moonlighting functions as a scaffolding protein that binds to both the cytoskeleton and the glycine and GABA type A receptors, and plays a major structural role in synaptic signaling in the central nervous system[7]. The plant homolog of gephyrin, Cnx1, has also been described to interact with the cytoskeleton[11]. These moonlighting properties of gephyrin homologs are an evolutionary trait thought to have been acquired in eukaryotes[1]. More recently, the discovery of a gephyrin-like protein (Glp) in bacteria[8] raised the question of whether the MoeA protein has the plasticity to adopt secondary functions and whether this plasticity is an ancient trait. *Corynebacteriales* contain two or more homologs of MoeA, one of them, Glp, plays an important role in cell division by directly binding the tubulin-like cytoskeletal protein FtsZ and an associated membrane protein GlpR[8]. Reminiscent of the eukaryotic gephyrin secondary function, Glp is thought to be similarly involved in network organization at the inner membrane of the corynebacterial septum[8]. Glp is phylogenetically distinct from *Escherichia coli* MoeA and seems to be the result of a duplication within the phylum Actinobacteria, and it is currently unknown if Glp has retained its enzymatic function[8].

It is remarkable that proteins sharing a high degree of sequence and structural conservation can present such diverse molecular functions and be involved in very distinct cellular mechanisms, including neurotransmission, cell division, and enzymatic catalysis. Moreover, it is striking that this

[1]Institut Pasteur, CNRS UMR 3528, Université Paris Cité, Structural Microbiology Unit, F-75015 Paris, France. [2]Institut Pasteur de Montevideo, Bioinformatics Unit, 11200 Montevideo, Uruguay. [3]Institut Pasteur, CNRS UMR 3528, Université Paris Cité, Bacterial Cell Cycle Mechanisms Unit, F-75015 Paris, France. ✉e-mail: dmegrian@pasteur.edu.uy; anne-marie.wehenkel@pasteur.fr

phenomenon occurred at least twice during the course of the evolutionary history of the species.

The seeming functional plasticity of MoeA raises several evolutionary questions. First, it is not clear how and when the protein fusion occurred during the evolutionary history of Eukaryotes, as this event has been studied independently only in a few plant, fungi and animal model organisms[1]. While this fusion is essential for the binding to - and network organization of - neurotransmitter receptors[12], it is not clear how widespread the fusion is in the tree of eukaryotes and what the effect of the fusion was in eukaryotes that do not have a nervous system. Moreover, it has not been reported how and when eukaryotes acquired MoeA. Besides eukaryotes, MoeA is ubiquitously present in Bacteria and Archaea[13], but the functionality and the evolutionary history of this protein in the context of the tree of life has not been addressed in detail and the recent discovery of a gephyrin-like protein in bacteria[8] suggests that functional repurposing of the MoeA scaffold is not a unique feature of higher eukaryotes. To understand the evolutionary history of MoeA, we present here the phylogenetic analysis of MoeA in all domains of life and show that both Archaea and Actinobacteria have independently undergone gene duplications and stably maintained two distinct clades over time. We show that the unique MoeA/gephyrin copy of Eukaryotes has a bacterial origin, and that the MogA-MoeA fusion occurred at least twice during the evolution of the Eukaryotes. Finally, we address the question on whether the MoeA homologs found in organisms that have more than one copy could be moonlighting proteins or repurposed enzymes that have lost their original catalytic function. Our analysis shows that the bacterial Glp homologs have a greatly altered active site in line with a possible loss of function, whereas in Archaea both copies seem to have a conserved active site, but possibly different substrate affinity. Our combined phylogenetic and structural analyses emphasize the functional differences between MoeA homologs, and leads us to propose a scenario for the diversity and evolution of this protein.

## Results

### Overall distribution of MoeA and domain architecture in eukaryotes

The complex chemical transformations required for the biosynthesis of Moco are strictly conserved throughout all domains of life. To be catalytically active, molybdenum is scaffolded with a molybdopterin containing pterin (MPT) to form Moco[6]. The biosynthesis of Moco comprises three major chemical rearrangements: (i) the circularization of GTP into cPMP, (ii) the transfer of sulfur to cPMP to generate MPT, and (iii) the insertion of molybdate into MPT to form Moco (Fig. 1a)[6]. These reactions are all catalyzed by highly conserved enzymes that occur individually (prokaryotes) or as multi-enzyme fusion proteins (eukaryotes)[6] (Fig. 1b, c). In prokaryotes, each step is catalyzed by the dual action of two individual proteins (Fig. 1a)[6]. In higher eukaryotes these pairs of proteins are fused to form 3 multi-domain proteins: MOCS1, MOCS2 and gephyrin (Fig. 1b)[6]. Plants represent an intermediate case, where only proteins MogA and MoeA are fused into the multi-domain gephyrin-homolog Cnx1[6]. Eukaryotic gephyrin contains two globular domains, G and E, respectively homologous to prokaryotic proteins MogA and MoeA, connected through a disordered linker region called C-domain that interacts with microtubules (Fig. 1c)[12]. Plant Cnx1 has a shorter C-domain, which may affect its quaternary structure, and has also been described to interact with the cytoskeleton, but in this case through its G-domain[14].

To infer the origin of MoeA in Eukaryotes we reconstructed a phylogeny including sequences obtained from the three domains of life (Fig. 1d and Supplementary Fig. 1). This phylogeny robustly places the Eukaryotes as a monophyletic group within Bacteria, suggesting that MoeA was acquired from Bacteria by the last eukaryotic common ancestor (LECA) or very early during the evolution of the Eukaryotes. The most evident difference between bacterial MoeA and animal gephyrin is the presence of the fused MogA (G-domain) to the N-ter of gephyrin[12]. This MogA-MoeA fusion is thought to have endowed gephyrin with its networking properties, as it allows for G-domain trimerization coupled to E-domain dimerization[15]. To understand how and when the transition between MoeA and gephyrin happened during

evolution, we investigated the domain distribution of MoeA proteins in Eukaryotes. Eukaryotic MoeA can be clustered roughly in two big groups: a first group belonging to algae, plants, and microbial eukaryotes (the Sar supergroup), and a second group belonging to protist clades Amoebozoa and Discoba, and the opisthokonts, which include fungi and animals[16] (Fig. 1e and Supplementary Fig. 2). Members of the first group have either the canonical MoeA architecture or an extra MogA domain, but in contrast to gephyrin, this domain is fused to the C-terminus of the protein. The reconstruction of a MogA domain phylogeny did not provide enough signal to infer if the fusions happened independently during evolution, or if it happened once and the MogA domain was relocated later. The extra MogA domain is present in most phyla of the Sar supergroup and Embryophyta (plants), but it is absent from algae and Sar phyla Pelagophyceae and Bacillariophyta. It is unclear whether the MoeA-MogA fusion in this group is ancestral and was lost later in some species, or alternatively whether the fusion happened at least three times during the evolution of these lineages.

A significant change happened in the ancestor of Amoebozoa, Discoba and the opisthokonts, as most members contain a MogA domain fused at the N-terminus of the MoeA domain (Fig. 1e and Supplementary Fig. 2). This fusion was an essential step for the transition between the canonical MoeA responsible for Moco biosynthesis, and the moonlighting gephyrin that is also responsible for postsynaptic clustering of neurotransmitter receptors in animals, as the N-terminal MogA domain is present in all studied animals. It is interesting to note that in fungi, MoeA was either kept in this fused form or was completely lost from the genome, as in the model organism *Saccharomyces cerevisiae* (Supplementary Figs. 2 and 3). We looked for MoeA homologs in representative members of all fungal orders and found that MoeA is missing in all 19 analyzed genomes of Microsporidia and Cryptomycota, suggesting that it was lost in the ancestor of these groups (Supplementary Fig. 3). We further identified several, possibly independent, losses scattered in the reference tree of Fungi, including members of Chytridiomycota, Blastocladiomycota, Zoomycota, Basidiomycota and Ascomycota (Supplementary Fig. 3). It is not clear why and how several fungi could circumvent independently the need for the Moco cofactor.

Taken together these results show that Eukaryotic MoeA has a bacterial origin, and that the transition to moonlighting gephyrin involved changes in its domain architecture, in particular the fusion of the MogA domain to the N-terminus of MoeA.

### Most Archaea contain two copies of MoeA from an ancestral MoeA duplication

Moco biosynthesis is widely conserved in Archaea and MoeA has been reported to be duplicated in some organisms like *Pyrococcus furiosus*[17]. Most Archaea are believed to prefer tungsten over molybdenum for the metal cofactor biosynthesis[18], but it is not clear whether the MoeA duplication is related to this variation. To understand the evolutionary history of MoeA in the prokaryotic context, we carried out a phylogenetic analysis of all Archaea and Bacteria. We identified MoeA in all archaeal phyla except for most members of Methanomassiliicoccales, Aciduliprofundales and Poseidoniales -all of which belong to the candidate phyla Thermoplasmatota- and the members of DPANN superphyla (Fig. 2a and Supplementary Data 1). The members of the DPANN are also known as nanoarchaea because of their greatly reduced cellular and genomic size. They live as epibionts and may have lost Moco biosynthesis enzymes because they can obtain the cofactor from the host[19]. Interestingly, most Archaea have two copies of MoeA, and some members of Methanomicrobiales have up to six copies, obtained by independent and recent duplications. The phylogeny of MoeA shows two well-supported subtrees (MoeA1 and MoeA2) that contain most archaeal phyla and present a topology that roughly matches that of the species tree of Archaea (Fig. 2b and Supplementary Fig. 4). This indicates that MoeA was duplicated before the divergence of Archaea -or early during its evolution- and the two paralogs were maintained in most phyla, suggesting an important functional role for both copies. In most cases, the two paralogs are contiguous in the genome, suggesting their participation in related functions (Fig. 2c).

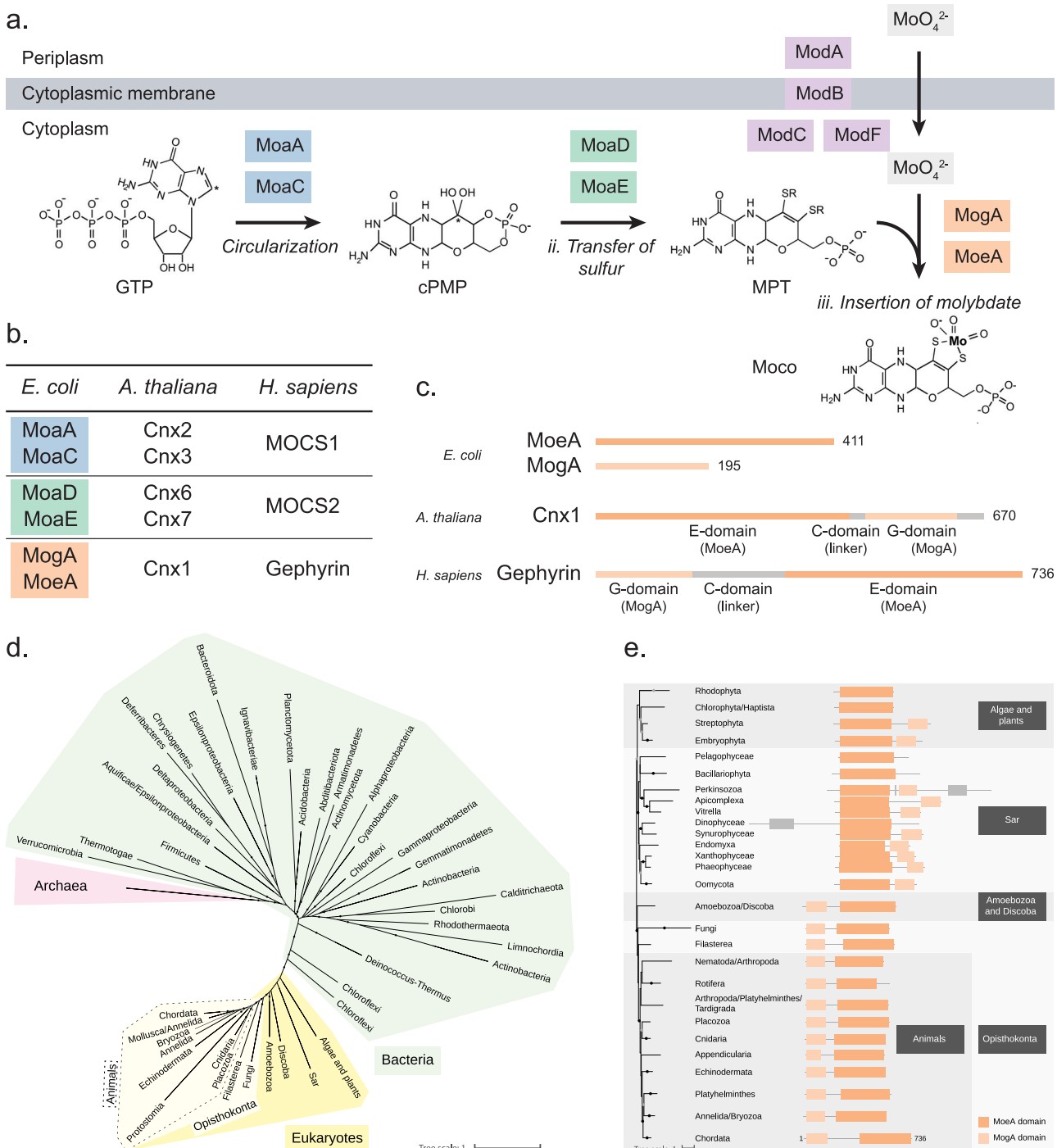

**Fig. 1 | Proteins involved in the biosynthesis of Moco. a** Schematic representation of the steps involved in the entrance of molybdenum ($MoO_4^{2-}$) to the cell, and in the biosynthesis of Moco in *E. coli*. Protein names are indicated in colored boxes. **b** Comparison of proteins involved in Moco biosynthesis in a representative species of Bacteria (*E. coli*), plants (*A. thaliana*) and animals (*H. sapiens*). **c** Detail of the organization of MoeA and MogA domains in a representative species of Bacteria (*E. coli*), plants (*A. thaliana*) and animals (*H. sapiens*). Each line corresponds to an individual protein. Numbers indicate the length of the protein. Domain MoeA is indicated in dark orange, and domain MogA in light orange. **d** Maximum-likelihood phylogeny of MoeA/Gephyrin in Eukaryotes, Bacteria and Archaea. Monophyletic groups were collapsed into a single branch. Note that the topology of the Eukaryotes subtree agrees with that of the species phylogeny[16,48]. Black dots indicate Ultrafast Bootstrap supports (UFB) > 90, gray dots indicate 80 < UFB < = 90 and branches without dots indicated UFB < = 80. The scale bar represents the average number of substitutions per site. For the detailed tree, see Supplementary Fig. 1. **e** Domain organization of MoeA/Gephyrin in representative species of Eukaryotes. Domains are mapped on the Eukaryotes MoeA/Gephyrin protein phylogeny. Domains indicated in gray correspond to domains different to MoeA or MogA. Higher taxonomic ranks are indicated in black boxes. The Sar clade includes stramenopiles, alveolates, and Rhizaria. The scale bar represents the average number of substitutions per site. For the detailed tree, see Supplementary Fig. 2.

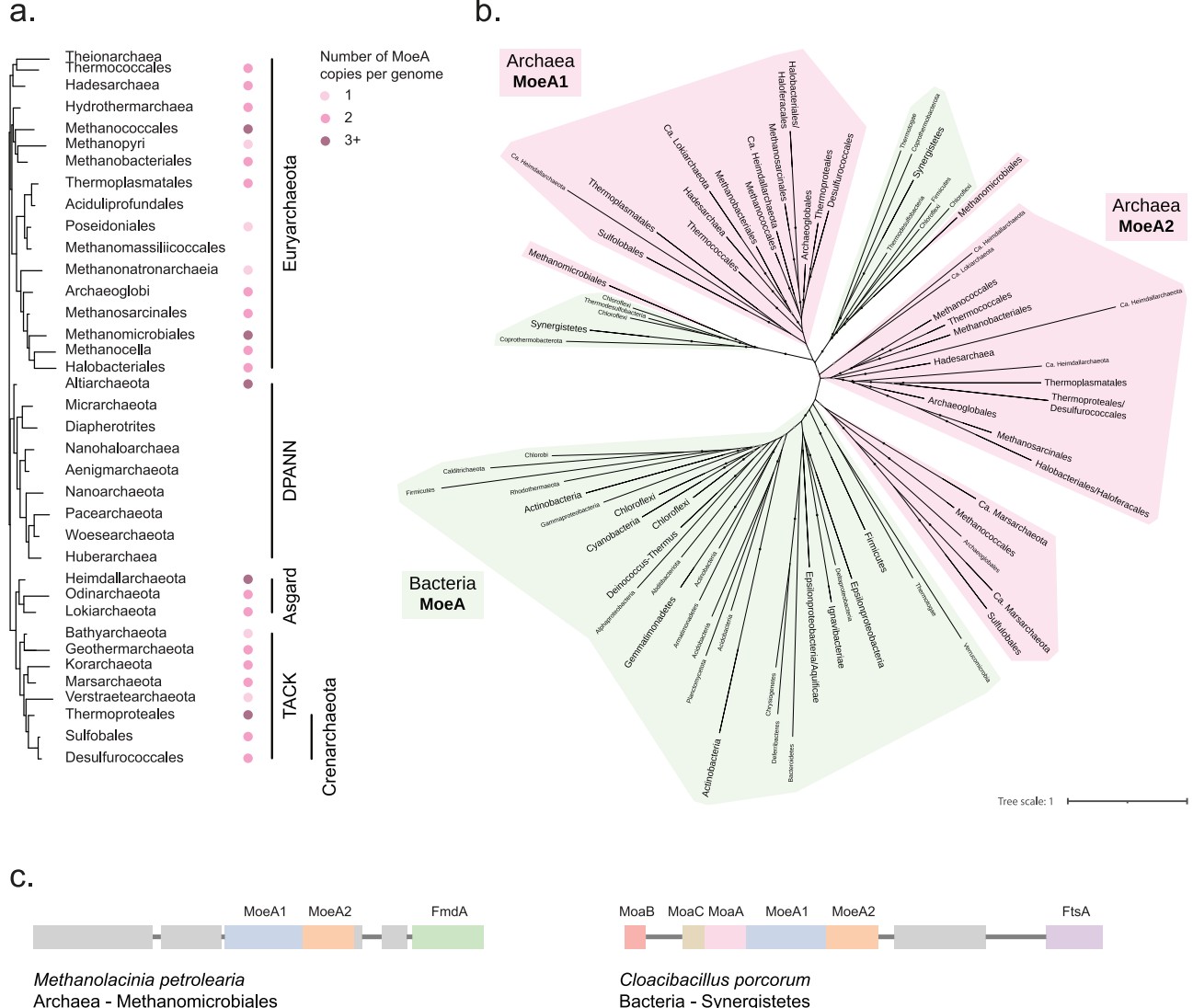

**Fig. 2 | MoeA distribution in Archaea and Bacteria. a** Phyletic pattern of the presence of MoeA in Archaea. Higher taxonomic ranks are indicated on the right. **b** Maximum-likelihood phylogeny of MoeA in Archaea and Bacteria. Monophyletic groups were collapsed into a single branch. Labels of branches that correspond to a collapsed group have bigger fonts than branches that correspond to single sequences.

Bacteria phyla are indicated in green, and Archaea phyla are indicated in pink. Black dots indicate UFB > 90, gray dots indicate 80 < UFB < = 90 and branches without dots indicated UFB < = 80. The scale bar represents the average number of substitutions per site. For the detailed tree, see Supplementary Fig. 4. **c** Genomic context of a representative archaeal MoeA and a bacterial MoeA that branch within Archaea.

We found two small bacterial clades branching within the clade formed by archaeal MoeA1 and MoeA2, both of which are phylogenetically distinct to the canonical bacterial MoeA clade that contains *E. coli* and most other bacterial species (Fig. 2b). These smaller clades contain the same species, which are anaerobic or facultative aerobic, meso- or thermophilic, and were sampled from a wastewater treatment plant (*Brevefilum fermentans*)[20], a hot spring sulfur-turf (*Caldilinea aerophila*)[21], swine intestinal tract (*Cloacibacillus porcorum*)[22], a methanogenic reactor treating protein-rich wastewater (*Coprothermobacter platensis*)[23], a methanogenic sludge (*Thermanaerovibrio acidaminovorans*)[24] and hot aquatic environments (*Thermodesulfobacterium commune*)[25]. The sister groups to both small bacterial clades correspond to the Methanomicrobiales (Fig. 2b), an order of anaerobic archaea that produce methane and inhabit aquatic sediments, anaerobic sewage digestors and the gastrointestinal tract of animals[26]. This suggests that the Methanomicrobiales likely coexist or coexisted in the same environment with these bacteria. While it cannot be excluded that the two small bacterial clades correspond to an ancestral duplication, the fact that both homologs are contiguous in the genome (Fig. 2c), that they are phylogenetically distinct to the canonical bacterial MoeA, and that the species containing these

homologs inhabit the same niches as Methanomicrobiales, suggests that these bacterial species could have obtained both MoeA copies from Methanomicrobiales in a single horizontal gene transfer (HGT) event. The bacterial acquisition of these archaeal proteins might have given them the ability to incorporate tungsten, instead of, or in addition to, molybdenum in the metal cofactor, which is rare in Bacteria[3]. The placement of the Methanomicrobiales clades in the tree of Archaea is intriguing, as we would have expected it to branch together with Methanosarcinales[27]. The support of the deepest branches in the phylogeny does not allow us to determine if this corresponds to a different evolutionary history of the MoeA copies of the Methanomicrobiales, or to an artifact, likely caused by long branch attraction (Fig. 2b).

### One of the archaeal paralogs is fused to a PBP domain and potentially binds tungsten

To understand the differences between the two MoeA paralogs in Archaea and to look for possible alternative functions of MoeA, we compared the sequences of the two archaeal paralogs. Most sequences in the MoeA1 clade are longer than the canonical *E. coli* MoeA (Fig. 3a and Supplementary

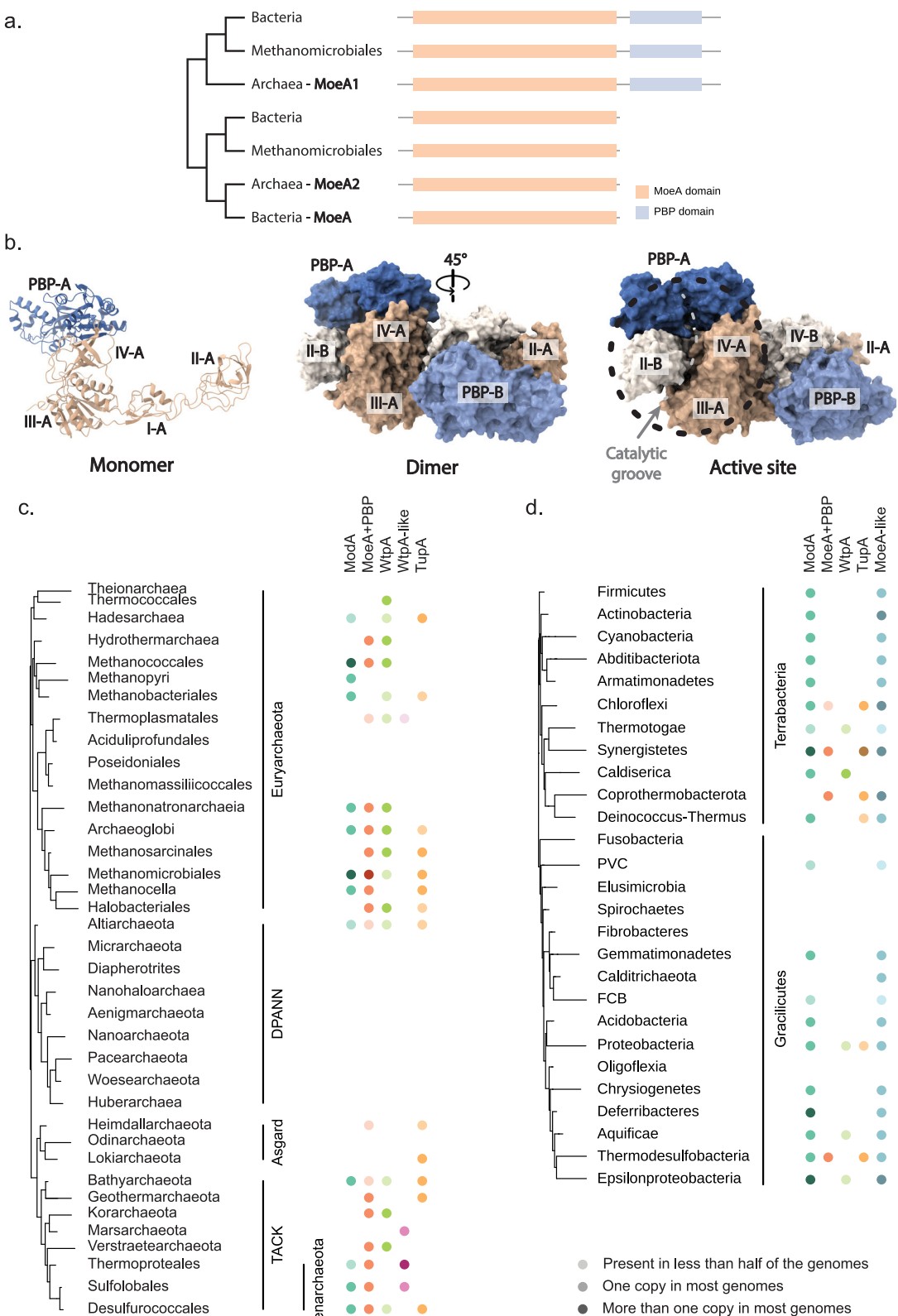

**Fig. 3 | PBP-like domain fusion to archaeal MoeA1. a** Schematic representation of the domain organization of MoeA in Archaea and Bacteria mapped on a schematic tree based on the phylogeny in Fig. 2b. For the detailed information, see Supplementary Fig. 4. **b** Alphafold protein structure of the MoeA1 dimer of archaeon *Archaeoglobus fulgidus*. Monomer A is indicated in darker shades than monomer B.

MoeA domains are indicated in shades of orange, and the PBP domain is indicated in shades of blue. **c** Phyletic pattern of the presence of molybdenum/tungsten related PBP proteins in Archaea. For the detailed information, see Supplementary Data 2. **d** Phyletic pattern of the presence of molybdenum/tungsten related PBP proteins in Bacteria. For the detailed information, see Supplementary Data 2.

Fig. 4). The analysis of these sequences shows the fusion of MoeA with a periplasmic-binding protein-like (PBP-like) domain in the C-terminal region (Fig. 3a and Supplementary Fig. 4). PBPs are nonenzymatic receptors used by prokaryotes to sense small molecules in the periplasm and transport them into the cytoplasm via ABC transporters[28]. Interestingly, the molybdate binding lipoprotein ModA involved in molybdenum uptake is a PBP[18]. Molybdate uptake systems have been mostly studied in bacteria, where they consist of a three-protein machine encoded by the modABC cassette, in which ModA is a molybdate binding lipoprotein, ModB an integral membrane protein and ModC an ATP-binding cassette ABC-type transporter (Fig. 1a)[18].

In bacteria, ModA mediates the entrance of molybdenum into the cell, where it is incorporated into MPT-AMP by MoeA to form the Moco (Fig. 1a)[29]. The archaeal PBP fusion to MoeA is likely cytoplasmic, as the canonical MoeA protein is found intracellularly, and there are no predicted signal peptides, that would suggest export into the periplasm, or predicted transmembrane domains between MoeA and the PBP domain that would suggest a communication through the membrane. The high-confidence AlphaFold atomic model of the MoeA-PBP dimer shows that the PBP-like domain sits on top of domain IV of MoeA (Fig. 3b). This relative positioning allows for the formation of a continuous groove between the predicted ligand-binding site of the PBP and the MoeA active site (Fig. 3b), suggesting that the PBP-like domain could facilitate the capture and channeling of the solute (molybdenum, tungsten, or other small molecules) into the active site of the dimer. To better characterize this PBP domain, we compared and investigated the presence or absence of other PBP proteins involved in the uptake of molybdenum and tungsten in Archaea: ModA (molybdenum and tungsten), WtpA (molybdenum and tungsten), and TupA (tungsten specific)[18]. We did not identify any of these proteins in Aciduliprofundales, Poseidoniales, Methanomassiliicoccales and the DPANN superphylum, in agreement with the absence of MoeA (Fig. 3c and Supplementary Data 2). All ModA, WtpA, TupA and the PBP domain of MoeA1 are widely distributed in Archaea, and it is not clear if they have the same function but are regulated differently, or if they evolved specialized functions.

Interestingly, the PBP domain of the archaeal MoeA1 family is also found in one of the bacterial MoeA homologs that branches within Archaea (Fig. 3a). While ModA is present in all bacterial phyla that contain a MoeA-like protein, the presence of the MoeA-PBP fusion protein as well as the tungsten transporters WtpA and TupA are restricted to a few phyla (Fig. 3d and Supplementary Data 2). Surprisingly, when MoeA-PBP is present, it cooccurs with the tungsten transporter TupA in 86% of the cases, and in 28% of the cases the coding sequences are located in the same locus (Supplementary Fig. 5 and Supplementary Data 3). This putative functional link between MoeA-PBP and TupA, suggests that MoeA-PBP might insert tungsten instead of molybdenum into MPT and might thus be involved in the biosynthesis of the tungsten cofactor instead of Moco. Considering the putative role of MoeA-PBP in the insertion of tungsten into MPT and the fact that the bacterial MoeA-PBP is phylogenetically related to the archaeal MoeA1 (Fig. 3a), this would suggest that the archaeal MoeA1 is specialized in the biosynthesis of the tungsten cofactor in Archaea, while MoeA2 takes part in the biosynthesis of the molybdenum cofactor. The fact that the two copies of MoeA in Archaea are ancestral and widespread in the domain, might suggest an important role for both cofactors in the metabolism of Archaea.

## Moonlighting, repurposing and specialization of MoeA

The functional canonical MoeA assembly is a homodimer that contains two catalytic sites with molybdopterin transferase enzymatic activity (Fig. 4a). In animals, MoeA is a moonlighting protein, as the unique copy of the gene in the genomes carries both the canonical molybdopterin transferase enzymatic activity and has acquired additional functions in protein network organization at the post-synapse[12]. Interestingly, we identified at least two copies of MoeA in Archaea and in some Bacteria (Fig. 2a, Supplementary Fig. 4 and Source Data), which may reflect specific physiological needs. For

instance, Actinobacteria have systematically maintained two or more MoeA copies throughout their evolutionary history[8]. MoeA duplications could thus indicate functional redundancy, or alternatively, different paralogs could have evolved different functions. Indeed, protein redundancy conserved over very large evolutionary distances in prokaryotes seems unlikely as the evolution of genomes appears to be dominated by reduction, and duplicated genes become either specialized or are lost[30]. Although the four domains of the MoeA monomeric structure are highly similar between all MoeA proteins, a conformational change between the domains leads to an important difference in the quaternary organization in the bacterial Glp homologs[8]. This conformational change between the domains translates into an opening of the dimer interface of Glp, that in turn generates the binding site for FtsZ and possibly GlpR[8] (Fig. 4a).

To explore the functional divergence or the possible dual role of all MoeA homologs in all the domains of life we computed AlphaFold high confidence structural models for each of the main MoeA groups (archaeal MoeA1 and MoeA2, bacterial MoeA and Glp, and eukaryotic MoeA and gephyrin), and mapped the sequence conservation onto representative models (Fig. 4a and Supplementary Fig. 6). The predicted structures of MoeA revealed that each group has the same overall monomeric and dimeric structures (Fig. 4a). In all homologs from Eukaryotes and Archaea, we observed a clear sequence conservation of the two protein regions that define the catalytic groove in the MoeA dimer, suggesting that they have a functional active site (Fig. 4a). Interestingly, the quaternary organization of the archaeal MoeA1 shows an opening at the dimer interface, which might reflect a specialization of the protein. On the contrary, the putative active site of bacterial Glp homologs showed a low degree of conservation (Fig. 4a and Supplementary Fig. 6), strongly suggesting the loss of the Moco biosynthesis capability, in line with a scenario of evolutionary repurposing rather than moonlighting.

To understand whether quaternary structural rearrangements in the MoeA dimer could reflect a functional conservation or divergence, we compared the distances defined by the amino acids in the active site of representative structures of the different MoeA groups. For quantification purposes, we chose conserved representative amino acids from the active site, computed their relative distances, and performed a Principal Component Analysis (PCA) to identify if the distances between the residues in these regions can discriminate the different MoeA groups (Fig. 4b and Supplementary Data 4). The PCA analysis based on the distances calculated for the active site separates Glp in the PC1 axis and archaeal MoeA1 in the PC2 axis from the other MoeA groups that cluster together (Fig. 4b). The fact that Glp representatives are spread in the PC1 axis suggest a loss in the conservation of the active site structure, which is congruent with the lack of sequence conservation, supporting the hypothesis of the loss the Moco biosynthesis capability. On the other hand, the separation of archaeal MoeA1 suggests that the distances in the active site of MoeA1 are conserved but are different to the distances in the other groups (Fig. 4b). This could be the consequence of the putative specialization of MoeA1 to bind tungsten instead of molybdenum, and/or the consequence of the physical constraints determined by the fusion of the PBP-like domain that sits on top of the active site.

Finally, the gephyrin homodimer contains the two catalytic sites with molybdopterin transferase enzymatic activity, as well as two binding sites for the GlyR and GABAA membrane neuroreceptors. To understand when during evolution MoeA acquired the potential to bind to these membrane receptors we analyzed the protein sequence conservation on the key binding residues (Fig. 5). The neuroreceptor binding site in gephyrin is very well conserved, however, this region is also well conserved in MoeA belonging to Sar, algae, and plants (Fig. 5), organisms that lack a nervous system. This conservation is absent in the bacterial MoeA, indicating that it appeared in the LECA or very early during the evolution of the Eukaryotes.

This result suggests that MoeA of Sar, algae, and plants might be able to bind other similar molecules in the same position, potentially an ancestral receptor, granting the non-animal eukaryotic MoeA another moonlighting function.

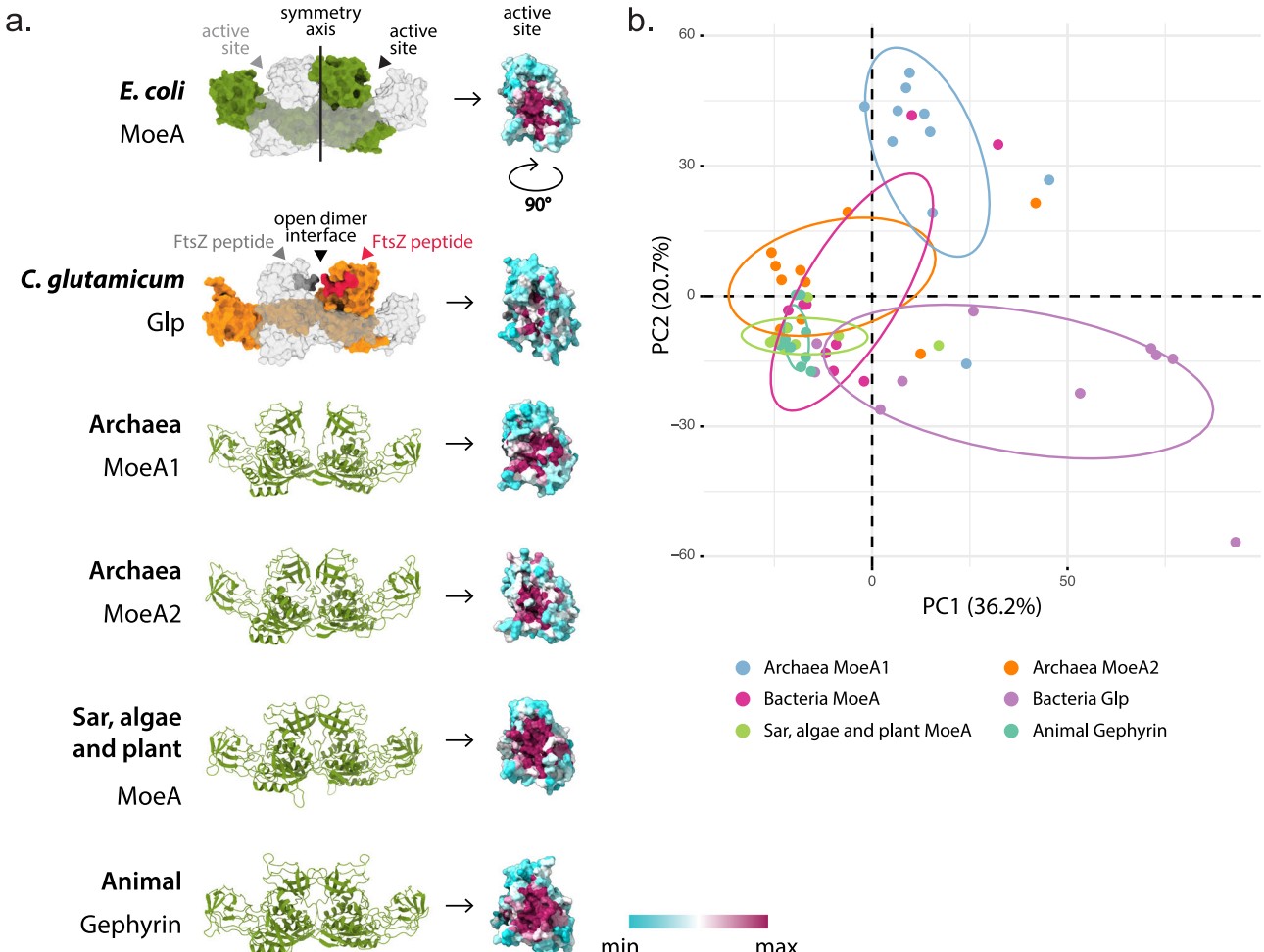

**Fig. 4 | Conservation analysis of the MoeA active site. a** On the left, the high-confidence AlphaFold atomic model of the MoeA dimer of representatives of all domains of life. The two symmetric active sites are indicated on the *E. coli* structure. Below, the Glp dimer of *C. glutamicum* (PDB id 8bvf), indicating the FtsZ binding sites. On the right, the sequence conservation of the MoeA active site mapped on a representative structure of each group. **b** Plot of the first two components from the PCA analysis of the distances between residues involved in the active site of MoeA in the different taxonomic groups. Each colored dot represents a linear combination of the distances between all the residues in the active site for an individual MoeA protein structure. For the detailed list of the distances, see Supplementary Data 4.

## Discussion

Our results support the hypothesis that MoeA was present in the last universal common ancestor (LUCA), which suggests that LUCA had pterin-based cofactors, as it has been proposed before[2]. Our results indicate that during the evolution of life, MoeA was transmitted mostly vertically, but it also underwent duplications, horizontal gene transfers and fusions, which led to the repurposing, acquisition of moonlighting function and probably specializations of the protein (Fig. 6). The most ancestral event we predict was a duplication of MoeA, which is reflected by the presence of two MoeA copies in most archaeal genomes, which form two separate clades in the phylogeny of MoeA (Figs. 2 and 6). These clades follow roughly the archaeal species tree[27], suggesting that the two copies were inherited vertically in most archaea. The presence of two MoeA copies was reported before in some archaeal species[17].

The history of MoeA in Bacteria seems less straightforward. Besides the largest clade, we identified two smaller clades of bacterial MoeA within methanogenic archaea (Figs. 2 and 6). Each of these clades can be evolutionarily associated to either archaeal MoeA1 or MoeA2 based on the length, domain architecture and tridimensional structure. However, it is not clear how bacteria obtained them. The topology of the phylogeny is compatible with two scenarios: either the duplication of MoeA happened before LUCA and the last bacterial common ancestor (LBCA) had two ancestral MoeA copies that were later lost in most bacteria, or the

LBCA had a single MoeA copy and some lineages acquired the archaeal MoeAs by HGT. The bacteria identified in these clades coexist with archaea in the same thermophilic and methanogenic niches[20–26], which supports both options. In the first scenario, both MoeA copies were obtained vertically by archaea and bacteria from LUCA. Bacteria that colonized other niches lost these genes, while a new MoeA could have been acquired from Archaea, and was later spread in the bacterial domain. The topology of the largest bacterial MoeA clade is not compatible with the bacterial species tree[31], which might suggest that MoeA was spread in Bacteria by HGT. However, the resolution of a single gene tree covering the two prokaryotic domains has important limitations, especially at the nodes that connect phyla[31] and can lead to misinterpretation of the events. In the second scenario, the LBCA had a single MoeA copy, and some bacterial species obtained, in a single event, the two archaeal MoeA genes by horizontal gene transfer from a methanogenic archaea. It is important to highlight that both archaeal MoeA genes, when present, are contiguous in the archaeal and bacterial genomes, which is compatible with a single HGT event.

Independently on how these two MoeA genes were obtained, it seems that both have an important and non-redundant role, at least in Archaea, as both were kept during billions of years of evolution. The fact that MoeA has been reported to utilize tungsten as well as molybdenum, and the existence of homologous enzymes that can use the tungsten

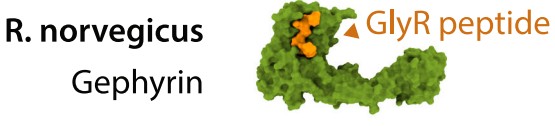

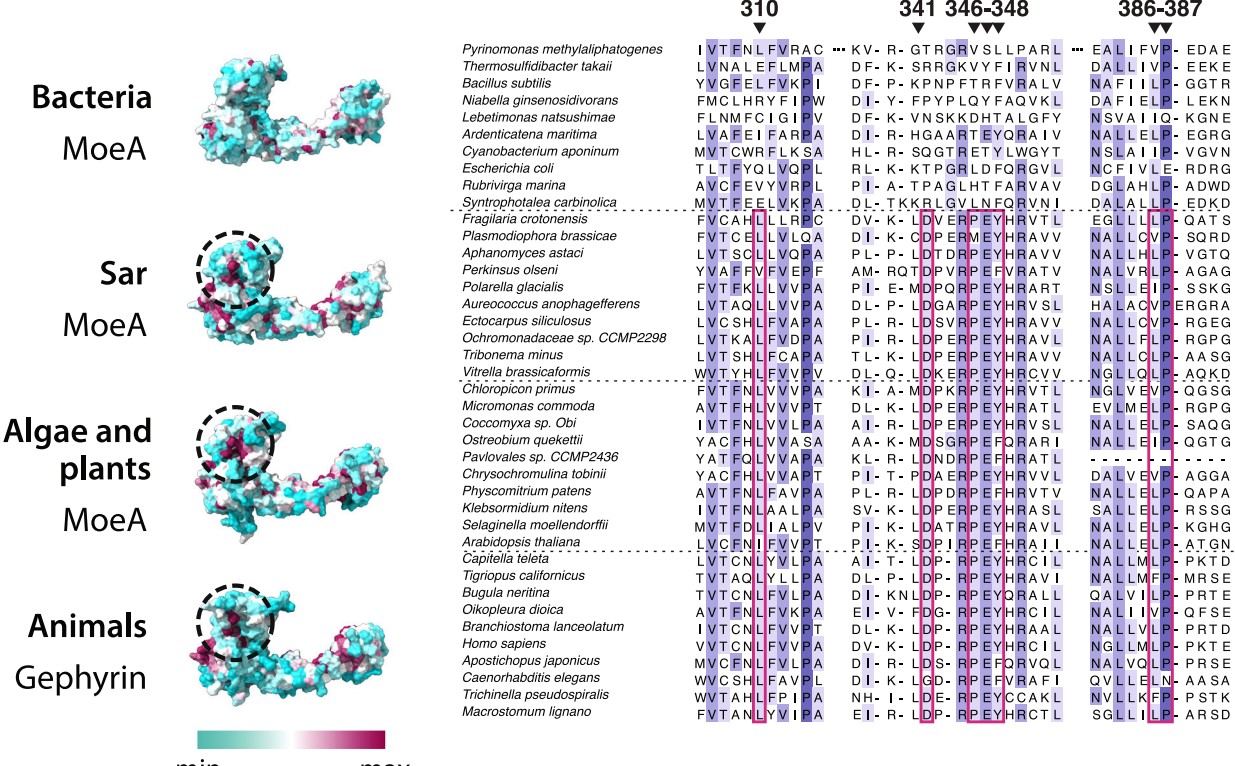

**Fig. 5 | GlyR binding site conservation in Eukaryotes.** On top, the protein structure of *Rattus norvegicus* gephyrin bound to a GlyR peptide (PDB id 4pd1) Below, sequence conservation of the MoeA membrane receptor binding site (dotted circle) mapped on a representative structure of each group indicated on the left. On the right, multiple sequence alignment of three fragments that form the binding site of MoeA/gephyrin that are conserved in most Eukaryotes, but not conserved in Bacteria. Key residues are indicated in red. All positions reported refer to the equivalent positions on *E. coli* MoeA.

**Fig. 6 | Scenario for the evolutionary history of MoeA in all domains of life.** All evolutionary events inferred in this work were mapped on a schematic phylogenetic tree based on the tree in Supplementary Fig. 1.

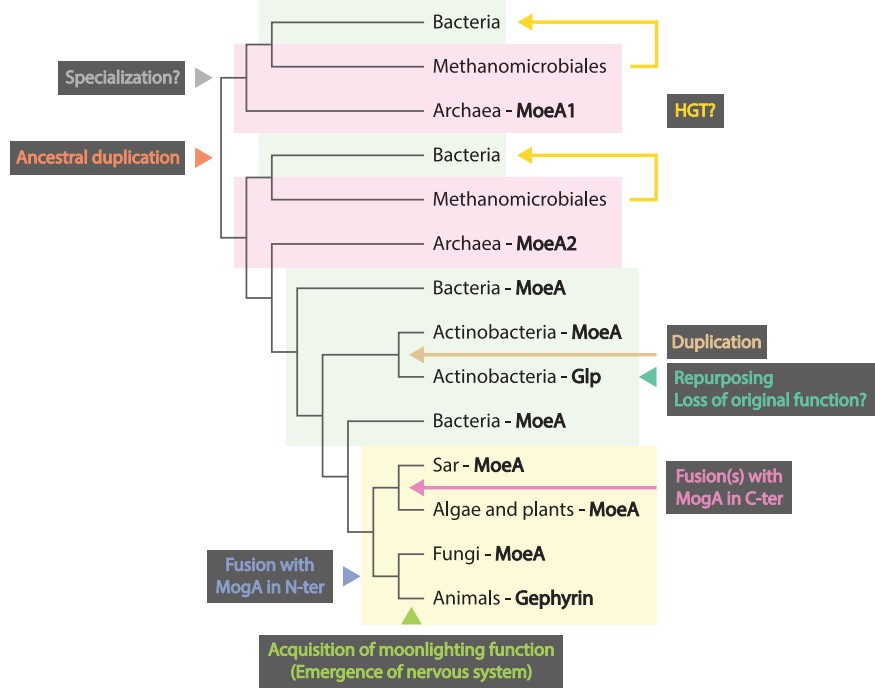

cofactor instead of Moco, leads us to put forward the hypothesis that one archaeal MoeA produces Moco, while the other produces a tungsten cofactor. The geological record suggests that tungsten was an essential element for the earliest life forms[32]. The ocean in the early Earth was anoxic and sulfidic, and under these conditions tungsten forms soluble salt while molybdenum is insoluble[32]. Around 2.5 billions years ago, the conditions of the ocean changed with the appearance of photo-synthesizing bacteria. This produced a rise of dioxygen in the environment, which oxidized molybdenum-containing sulfide minerals and led to the accumulation of molybdenum in the oceans. This event probably forced the metabolisms of cellular organisms to adapt to the changing conditions of the ocean, and to start using molybdenum instead of tungsten, by duplicating and maybe specializing the machinery involved in the biosynthesis of the pterin-based cofactor and the enzymes that use this cofactor[32]. Nowadays, tungsten is mainly used by thermophilic anaerobic archaea[32], whose anoxic environments have higher tungsten than molybdenum bioavailability. However, these organisms, as well as most archaea and some bacteria have both MoeA1 and MoeA2, and the functional differences between them are still not clear.

It had been reported that in bacteria, molybdate is mainly taken up by the ModABC system, however, ModA can bind both molybdenum and tungsten[18]. Also, a homologous molybdenum and tungsten trans-porter, WtpABC, and a third tungsten-specific transporter, TupABC, have been identified, but their distribution was reported to be much more restricted[18]. In this work we confirmed that ModA is widespread in almost all bacteria, and that the presence of WtpA and TupA is scattered in the phylogeny of bacteria. Interestingly, we identified a ModA homolog fused to MoeA in some bacteria. This MoeA, which is the homologous to archaeal MoeA1, co-occur with transporter TupA, sug-gesting a functional and evolutionary link. As TupA is a tungsten-specific transporter in Archaea, this result suggests that MoeA1 might be specific to tungsten and was either obtained or kept in bacteria that inhabit anoxic environments where molybdenum is less available than tungsten. In this regard, Mota et al.[33] studied the effects of molybdenum and tungstate on the expression levels of *moeA1* and *moeA2* of the bacterium *Desulfovibrio alaskensis*. The supplementation with tungsten did not affect the expression of *moeA2*, but decreased the expression of *moeA1*, while the supplementation with molybdenum did not affect the expression of both *moeA* genes. Using a different rationale, Malotky et al.[34] expressed MoeA1 and MoeA2 of the archaeon *Pyrococcus furiosus* in an *E. coli* MoeA mutant strain, and observed that MoeA2 partially complements the mutant, suggesting that archaeal MoeA2 has a similar function to bacterial MoeA. This result agrees with the topology of our MoeA phylogeny, that places the largest clade of bacterial MoeA closer to archaeal MoeA2 (Figs. 2 and 6), and supports the hypothesis that MoeA2 uses molybdenum.

We showed that MoeA was obtained by early eukaryotes from Bacteria (Figs. 1 and 6), and that during the diversification of the Eukaryotes MoeA fused to MogA in probably two separate events, once in the C-terminus, and once in the N-terminus. The MogA domain in plant Cnx1 is fused to the domain IV of MoeA, which is the one that binds to the neuroreceptors in Gephyrin. We could hypothesize that the relocation of MogA to the N-terminal region of Gephyrin set the domain IV free and permitted the gain of function at the synapses. Both types of fusion proteins have been reported to form networks and interact with the cytoskeleton, as it is the case of plant Cnx1 and animal gephyrin[11,12]. We recently reported a similar case in the Actinobacteria, where an independent duplication of MoeA within this phylum led to the spe-cialization of Glp, one of the MoeA paralogs[8]. This protein binds to the bacterial tubulin homolog FtsZ, and acts as a protein scaffold to control cell division and morphogenesis. Differently to gephyrin, in this work we predict that Glp does not have a moonlighting function, as the catalytic activity seems to have been lost during the specialization. Our results support the functional versatility and adaptive nature of the MoeA protein, which has been repurposed independently in both eukaryotes

and bacteria to carry out analogous functions in scaffolding and control at the inner membrane in dynamic systems, such as mammalian synaptic signaling and bacterial cell division. The potential of MoeA/gephyrin to create networks and to bind other proteins in eukaryotes that do not contain a nervous system, such as plants or fungi, reflected on their sequence conservation, opens up the question whether other cellular processes could be mediated by this versatile protein.

Overall, we propose an evolutionary scenario where MoeA was present in LUCA and is nowadays widespread in most species in all domains of life (Fig. 6). During its evolutionary history, MoeA was subjected to indepen-dent duplications (and possibly HGTs), that led to its specialization, repurposing and acquisition of a moonlighting function. Besides its meta-bolic role, MoeA seems to have acquired networking capabilities in an independent manner, probably favoring the acquisition of novel and diverging functions, as it is the case for actinobacterial Glp and animal gephyrin. It remains an open question whether other MoeA homologs have other specialized or moonlighting functions, and whether this versatility exists in other proteins that maintained the folding while changed or acquired new functions.

## Methods
### Database assembly
To carry out a large-scale MoeA investigation in all domains of life, we assembled databases with genomes representing all bacterial, archaeal and eukaryotic diversity. For Bacteria, we assembled a database containing 81 genomes (five taxa per phylum with cultured representatives), based on the taxonomic sampling in Martinez et al., 2023, and adding five actinobacterial taxa. For Archaea, we assembled a database containing 122 genomes representing all major phyla, based on the taxonomic sampling in ref. 35, but excluding the gen-omes that are not annotated in the NCBI Genome database[36]. For Eukaryotes, we selected five taxa per phylum (if available), from all eukaryotic annotated genomes in the NCBI Genome database[36]. We assembled a database containing the 129 genomes corresponding to these phyla, representing all diversity present at the NCBI as of October 2021. For Fungi, we assembled a database containing 171 genomes, including one representative of each fungal order with at least one annotated genome at the NCBI Genome database[36] as of October 2021.

### Homology searches and mapping
To study the taxonomic distribution of MoeA in all domains of life, we performed sensitive HMM similarity searches against the Bacteria, Archaea, Eukaryotes and Fungi databases. First, we built HMM profiles based on the bacterial MoeA alignment provided in ref. 8, using the HMMBUILD tool from the HMMER package[37]. Then, we used these profiles to search for MoeA homologs in the four databases, using the HMMSEARCH tool from the HMMER package[37] with by default parameters. To remove false posi-tives, we used the Conserved Domain Database (CDD) online tool[38] to identify hits that contain all three Pfam domains MoeA_N (pfam03453), MoCF_biosynth (pfam00994) and MoeA_C (pfam03454). We mapped the number of MoeA copies per archaeal genome on a schematic Archaea tree, obtained from[35] using iTOL[39]. To map the MoeA presence/absence in Fungi, first, we reconstructed a species Fungi phylogeny based on the DNA-directed RNA polymerase II subunit RPB2 protein. To identify RPB2 homologs in all fungal genomes, we used the JACKHMMER tool from the HMMER package[37] and the *Saccharomyces cerevisiae* NCBI RefSeq sequence (NP_014794.3) as the query, against the Fungi database, and we selected the best hit per genome. Complete absences of MoeA in any of the genomes in the four databases were manually verified.

To identify ModA, WtpA and TupA homologs in all domains of life, we used the JACKHMMER tool from the HMMER package[37], and the *Escherichia coli* ModA NCBI RefSeq sequence (NP_415284.1), *Pyrococcus furiosus* WtpA NCBI GenBank sequence (AAL80204.1), and *Campylo-bacter jejuni* TupA NCBI RefSeq sequence (YP_002344912.1) as the queries,

against the Bacteria and Archaea databases. We aligned the three groups of hits separately with MAFFT[40] using the L-INS-I algorithm, and we visually selected homolog sequences of each protein. We realigned these sequences, removed the columns with more than 20% of gaps, and built HMM profiles for each protein using the HMMBUILD tool from the HMMER package[37]. Then, we used these profiles to search for ModA, WtpA and TupA homologs in the Bacteria, Archaea and Eukaryotes databases using the HMMSEARCH tool from the HMMER package[37] with by default parameters, and we selected the hits with an e-value above $1e-^6$. We mapped the presence of these proteins on an Archaea[35] and a Bacteria[31] species phylogeny using iTOL[39].

To confirm the co-occurrence of TupA and PBP-MoeA, we performed a second search for these two proteins against all complete genomes of Bacteria deposited in the NCBI Genome database[36] as of October 2024 (3603 representative species). We mapped the co-occurrence of PBP-MoeA on a TupA phylogeny using iTOL[39].

### Phylogenetic analyses

We reconstructed three MoeA phylogenies including the homologs identified in: (i) Bacteria, Archaea and Eukaryotes, (ii) Eukaryotes, and (iii) Bacteria and Archaea; a species Fungi phylogeny, and a TupA phylogeny. To reconstruct these phylogenies, we aligned the protein sequences with MAFFT[40] using the L-INS-I algorithm, and we trimmed the alignment using trimAl[41], keeping the columns that contain less than 20% of gaps. We used these alignments to reconstruct a guide tree with IQ-TREE[42] using the Model Finder Plus (MFP) option. Then, we used these guide trees to reconstruct a maximum-likelihood trees with IQ-TREE[42] using the PMSF model, with Ultrafast Bootstrap supports calculated from 10.000 replicates, with a minimum correlation coefficient of 0.999.

We used the results of the CDD[36] described in the previous section to identify extra domains, like the PBP, in some MoeA homologs. We mapped the domain organization of MoeA into the (ii) Eukaryotes, and (iii) Bacteria and Archaea phylogenies using iTOL[39].

### Protein structure prediction and distance calculation

We predicted the structure of the dimeric form of ten representative MoeA homologs identified in all domains of life using AlphaFold[43]. To compare MoeA structures and based on the alignment of all MoeA homologs in all domains of life, we removed the N-terminal and C-terminal ends of each protein that do not align with *E. coli* MoeA (see Source Data). All positions reported on MoeA structures refer to the equivalent positions on *E. coli* MoeA based on the alignment, unless stated otherwise.

We classified MoeA structures into eight groups based on the phylogenetic analyses in the previous section: archaeal MoeA1, archaeal MoeA2, bacterial MoeA, bacterial Glp, Sar, a MoeA, Algae and plants MoeA, fungal MoeA, and animal Gephyrin. To map the sequence conservation on a representative structure of each group we used the MoeA alignments obtained in the previous section and software ChimeraX[44]. The method for calculating the sequence conservation is the entropy-based measure from software AL2CO[45]. For the list of representative structures see Supplementary Data 5.

To evaluate the conservation of the distances between residues in the active site of MoeA, we manually selected the residues on the active site surface (for details, see Supplementary Data 6). Then, we computed all distances between the residues in the active site for each predicted protein structure, using the Python Bio.PDB package[46]. Finally, we performed a PCA analysis to compare the distances between the residues of interest in the different MoeA groups.

### Statistics and reproducibility

The reliability of the phylogenetic trees was measured by the bootstrap probability of interior branches of the tree. We used the Ultrafast Bootstrap approximation implemented in IQ-TREE[42] with 1000 replicates.

## Data availability

All data supporting the findings of this study are available within the paper and its Supplementary Figures and Supplementary Data. Source Data and can be found in https://doi.org/10.17632/phw4knbn8m.2[47]. All other data are available from the corresponding author.

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

## Acknowledgements

This work was supported in part by grants from the Agence Nationale de la Recherche (ANR, France), contracts ANR-18-CE11-0017 (P.M.A.) and ANR-21-CE11-0003 (A.M.W.), and by institutional grants from the Institut Pasteur – including the Calmette-Yersin Internship Grant (D.M.) –, the CNRS, and Université Paris Cité. This work used the computational and storage services (maestro cluster) provided by the IT department at Institut Pasteur, Paris. Molecular graphics were done with ChimeraX, developed at UCSF with support from NIH (R01-GM129325) and NIAID. The authors would like to thank Martin Graña for his careful reading of the manuscript and his suggestions.

## Author contributions

D.M. and A.M.W. designed the research. D.M. carried out all bioinformatic analysis. D.M., M.M., A.M.W. and P.M.A. carried out the structural analysis. D.M. and A.M.W. wrote the paper. All authors edited the paper.

## Competing interests

The authors declare no competing interests.
