## [Transparent Peer Review file · Communications Biology]

Evolutionary plasticity and functional repurposing of the essential metabolic enzyme MoeA

Corresponding Author: Dr Anne Marie WEHENKEL

Version 0:

Reviewer comments:

Reviewer #1

(Remarks to the Author)

In the present manuscript Megrian and collaborators show the multifunctionality and evolutionary history of MoeA, an essential enzyme in molybdenum cofactor biosynthesis. They use phylogenetic inference and protein structure analyses to explore how MoeA has acquired additional roles beyond its enzymatic function. Key findings include a discussion of the gephyrin-like protein (Glp) in bacteria, which suggests ancient functional plasticity. The manuscript also identifies gene duplications in Archaea and Eukaryotes, leading to distinct evolutionary and functional adaptations.

Megrian and collaborators present a thorough and well-researched study on the evolutionary history and functional versatility of MoeA. The manuscript is well-written, with clear explanations of complex evolutionary concepts and detailed methodological descriptions. The integration of phylogenetic analyses with structural biology provides compelling evidence for the repurposing of MoeA in different domains of life. The research is significant and novel as it highlights the adaptive nature of enzymes and the evolutionary processes leading to multifunctionality. The broader implications for enzyme multifunctionality and adaptive potential may also be of interest from researchers studying protein evolution, functional genomics, and structural biology. By demonstrating the evolutionary plasticity of an essential metabolic enzyme, it challenges and expands current understanding of enzyme evolution and multifunctionality.

To enhance the overall quality and clarity of the manuscript, the following minor issues should be addressed:

1. Regarding the introduction, consider adding a bridging paragraph that succinctly connects the general background to the specific research questions addressed in this study.
2. Some figures should be provided in higher resolution: Fig 1a, Fig 4b.

I recommend the publication of this manuscript, provided the authors address the minor issues and suggestions above.

Reviewer #3

(Remarks to the Author)

In this article, Megrian and colleagues explore the functional diversification of the MoeA protein family using phylogenetics, structural and sequence analyses. The MoeA enzyme is part of the biosynthesis of the MOCO cofactor essential for the maturation of molybdenum-containing enzymes. Its particularity resides in its diversification that led to moonlighting properties: MoeA has supplementary additional functions in animals as it is involved in the organization of neurotransmitter receptors. The authors explore the diversity of the family across the tree of life, with a special emphasis on archaea, fungi and bacteria. Several evolutionary events are invoked, including fusion with the companion protein MogA twice in Eukaryotes, and duplications potentially leading to functional divergence and evolution of interaction with tungsten in Archaea. In support of phylogenetic analyses, the prediction and analyses of structures predicted with AlphaFold also enable the authors to study the evolution of some structural features in link with functional predictions.

Overall, the study is well led, and the paper is clearly written and reads well. The evolution of this ancient protein family that evolved moonlighting properties is very intriguing. The study can potentially interest researchers from different fields (molecular and cellular biology) and working with very different model organisms (from archaea to bacteria and eukaryotes). The functional interpretations remain speculative is the absence of experimental validation, especially for the archaeal

paralog that might interact with tungsten, but the hypotheses raised here are well-argued and could pave the way for future, more focused experimental investigations. Follow my more specific comments.

Major points

- 1) L. 55-56 – In the introduction, I think the role of the MOCO cofactor is not clearly described. There is potentially a missing sentence on how it is essential for many molybdoenzymes. Rather, the role of molybdenum is described.
- 2) Tree of MoeA on Fig. 1 and its evolutionary interpretation. Is that so clear whether the gene is mostly vertically transmitted or not? Along the same line, regarding line 139: is the MoeA tree rooted? If not, how can it be inferred that the Eukaryotes are a monophyletic group within Bacteria? In addition, is the Eukaryotes' subtree in agreement with the phylogeny of Eukaryotes? This would be a pre-requisite to suggest vertical transmission from LECA (or from another ancient ancestor). Those points should be further discussed to support the authors' claims.
- 3) How long is the MogA domain in the fused version of the protein in Eukaryotes? Would it be possible to build a MogA domain phylogeny? Are there "intact" copies of MogA left in Eukaryotes, or is the MogA domain the only one involved in MOCO biosynthesis with MoeA as suggested by Fig 1b? It is striking that the fusion of Mog1 in C-ter or in N-ter of MoeA likely occurred twice in Eukaryotes. Could the authors discuss the potential advantage of fusing the two proteins? Were there any cases observed in Prokaryotes?
- 4) L. 264-265. Could there be some statistics to support the claim that "MoeA-PBP and tungsten transporter TupA cooccur when present"? How many genomes are supporting this for each phylum? Is this significant? There seems to be two phyla where MoeA-PBP is absent while TupA is present. Could the authors rephrase the above sentence to make it more precise? i.e., that MoeA-PBP cooccur with tungsten transporter TupA when present?
- 5) I am not very convinced by the PCA analysis and its signification or significance, plus the colors used (and low resolution of the panel) makes it difficult to interpret. Could the authors elaborate on the reasons of their interpretation? And indicate the contribution of the different axes? Wasn't there another, maybe more straight-forward way to compare the potential active sites of the different proteins?

Minor points

- 6) Figure 1. The molecules' images could be sharper. In legend, please italicize the names of all the organisms. "Domain MoeA ..." the colors in the legend and on the figure do not match. Please also define "UFB" upon 1st appearance. It seems that the tree in panel (e) is not described. Is that still the tree of MoeA/Gephyrin? Or is that a species tree? Please define Sar in the legend.
- 7) In line 164, differential losses of MoeA are suggested in Fungi, but this does not show in Figure 1E. Is this the case for other lineages too?
- 8) Please write for Sup Fig. 3 that the red dot indicates the loss. Usually, it is the presence of a trait which is symbolized.
- 9) L. 171 and possibly fusion of MogA in C-ter? Are we sure that Cnx1-type proteins are not also moonlighting?
- 10) L. 307-308 are partially redundant with 302-304 concerning the archaeal homologs, the authors might want to rewrite this part a bit?
- 11) The resolution of the structures and of the PCA panel could be improved on Figure 4a.
- 12) L. 405. "evolutionary" instead of "evolutive"?
- 13) L. 451. Could the authors explain why there is so little bacterial genomes (81) compared to archaeal (122) and eukaryotic (129) ones? Whenever bacteria are more diverse? Is the dataset used here representative of the bacterial diversity?
- 14) L. 490. *E. coli* to be in italics. Same for *S. cerevisiae* in line 478 (plus other occurrences elsewhere).
- 15) In Methods, lines 504-510 are mostly repeating the phylogenetic reconstruction presented in lines 478-486. Could the authors rationalize the text? The fungi tree could probably be explained in the same section than that of the other trees, especially when the "mapping" part of MoeA and MoeA homologs is basically also the same.

Version 1:

Reviewer comments:

Reviewer #1

(Remarks to the Author)

The authors have adequately addressed all of my concerns from the initial review, incorporating the requested modifications into the revised manuscript. Based on these improvements, I am satisfied with the current version and recommend this work for acceptance and publication.

Reviewer #3

(Remarks to the Author)

I thank the authors for this revised version and the comprehensive responses to my questions/comments. I am fully satisfied with these and am happy to recommend this article for publication.

I only leave a few minor points/typos below, which I'm confident the authors will address.

- In Materials and Methods, line 464: please add how were selected the bacterial phyla, as you mentioned in the rebuttal that they are only representing the phyla with cultured representatives, and it is not indicated here where it should.
- Line 480: "sensitive homology searches": this is a shortcut. They are really "similarity searches", the goal being indeed by proxy to find homologues.
- Line 221: "fact that both homologs are continuous" => change to "contiguous"
- Line 357: "the species archaeal tree" => maybe change to "the archaeal species tree"
- Line 375: same here: "with the species bacterial tree" => change to "bacterial species tree"
- Line 494: there is a space missing.
- Some species names are still not in italics in Mat&Meth (e.g. L. 498)

Reviewers' comments:

Reviewer #1 (Remarks to the Author):

In the present manuscript Megrian and collaborators show the multifunctionality and evolutionary history of MoeA, an essential enzyme in molybdenum cofactor biosynthesis. They use phylogenetic inference and protein structure analyses to explore how MoeA has acquired additional roles beyond its enzymatic function. Key findings include a discussion of the gephyrin-like protein (Glp) in bacteria, which suggests ancient functional plasticity. The manuscript also identifies gene duplications in Archaea and Eukaryotes, leading to distinct evolutionary and functional adaptations.

Megrian and collaborators present a thorough and well-researched study on the evolutionary history and functional versatility of MoeA. The manuscript is well-written, with clear explanations of complex evolutionary concepts and detailed methodological descriptions. The integration of phylogenetic analyses with structural biology provides compelling evidence for the repurposing of MoeA in different domains of life. The research is significant and novel as it highlights the adaptive nature of enzymes and the evolutionary processes leading to multifunctionality. The broader implications for enzyme multifunctionality and adaptive potential may also be of interest from researchers studying protein evolution, functional genomics, and structural biology. By demonstrating the evolutionary plasticity of an essential metabolic enzyme, it challenges and expands current understanding of enzyme evolution and multifunctionality.

We would like to thank the reviewer for this positive assessment of our work.

To enhance the overall quality and clarity of the manuscript, the following minor issues should be addressed:

1. Regarding the introduction, consider adding a bridging paragraph that succinctly connects the general background to the specific research questions addressed in this study.

We have now reformulated this paragraph (lines 89-93)

2. Some figures should be provided in higher resolution: Fig 1a, Fig 4b.

We apologize for the low resolution of the images, we had to compress the files for the submission because of the file size limitations. We will make sure that the final versions of the figures are of high quality.

I recommend the publication of this manuscript, provided the authors address the minor issues and suggestions above.

Reviewer #2 (Remarks to the Author):

In this article, Megrian and colleagues explore the functional diversification of the MoeA protein family using phylogenetics, structural and sequence analyses. The MoeA enzyme is part of the biosynthesis of the MOCO cofactor essential for the maturation of molybdenum-containing enzymes. Its particularity resides in its diversification that led to moonlighting properties: MoeA has supplementary additional functions in animals as it is involved in the organization of neurotransmitter receptors. The authors explore the diversity of the family across the tree of life, with a special emphasis on archaea, fungi and bacteria. Several evolutionary events are invoked, including fusion with the companion protein MogA twice in Eukaryotes, and duplications potentially leading to functional divergence and evolution of interaction with tungsten in Archaea. In support of phylogenetic analyses, the prediction and analyses of structures predicted with AlphaFold also enable the authors to study the evolution of some structural features in link with functional predictions.

Overall, the study is well led, and the paper is clearly written and reads well. The evolution of this ancient protein family that evolved moonlighting properties is very intriguing. The study can potentially interest researchers from different fields (molecular and cellular biology) and working with very different model organisms (from archaea to bacteria and eukaryotes). The functional interpretations remain speculative is the absence of experimental validation, especially for the archaeal paralog that might interact with tungsten, but the hypotheses raised here are well-argued and could pave the way for future, more focused experimental investigations. Follow my more specific comments.

We would like to thank the reviewer for their positive evaluation of our work and agree that our work would gain in strength if we could experimentally prove some of our hypotheses. As this is beyond the scope of the current manuscript we sincerely hope that our work can serve as a basis to other researchers pursuing for instance tungsten uptake in Archaea.

Major points

1) L. 55-56 – In the introduction, I think the role of the MOCO cofactor is not clearly described. There is potentially a missing sentence on how it is essential for many molybdoenzymes. Rather, the role of molybdenum is described.

We have now reformulated this description (lines 52-55 and 65-66).

2) Tree of MoeA on Fig. 1 and its evolutionary interpretation. Is that so clear whether the gene is mostly vertically transmitted or not? Along the same line, regarding line 139: is the MoeA tree rooted? If not, how can it be inferred that the Eukaryotes are a monophyletic group within Bacteria? In addition, is the Eukaryotes' subtree in agreement with the phylogeny of Eukaryotes? This would be a pre-requisite to suggest vertical transmission from LECA (or from another ancient ancestor). Those points should be further discussed to support the authors' claims.

Reconstructing the evolutionary history of a single gene across the three domains of life is virtually impossible, as we do not know with confidence what the species tree of life looks like. We do not have an outgroup to root the tree (ideally, an ancient paralog or MoeA that diverged in LUCA), but considering the distribution of MoeA and the geological record, we speculate that the root should be placed around the branch that separates the two big clades of Archaea and Bacteria (therefore, LUCA). The resolution of the subtrees of the bacterial and archaeal domains does not allow us to confirm that the transmission within each domain was fully vertical, as we cannot reconstruct billions of years of evolution because the alignment relies on a very short sequence of only 300 positions. While we cannot reconstruct the deeper nodes with confidence, in Archaea, we can see some groupings of phyla that suggest that at least up to a certain level there was vertical inheritance (in Figure 2: MoeA2 (Halobacteriales, Methanosarcinales and Archaeoglobales; Thermoproteales and Desulfurococcales; Methanococcales, Thermococcales and Methanobacteriales; Lokiarchaea and Heimdallarchaea) and MoeA1 (Thermoproteales and Desulfurococcales; Methanosarcinales and Halobacteriales)). Regarding the Eukaryotes, we know for sure that MoeA was not originated in the Eukaryotes, so no matter where we put the root (any branch within Bacteria or Archaea), the Eukaryotes form a monophyletic clade within Bacteria. While the deepest nodes of the species tree of the Eukaryotes are still not resolved (before the appearance of Ophisthokonta), the overall topology of our MoeA subtree is in agreement with the currently accepted phylogenies of the Eukaryotes. We have now toned down the claim that MoeA was transmitted vertically in most species in all domains of life, and we have indicated that the Eukaryotes tree topology is in agreement with that of the species in the literature (lines 351-352; 587-588; 592-593).

3) How long is the MogA domain in the fused version of the protein in Eukaryotes? Would it be possible to build a MogA domain phylogeny? Are there “intact” copies of MogA left in Eukaryotes, or is the MogA domain the only one involved in MOCO biosynthesis with MoeA as suggested by Fig 1b? It is striking that the fusion of Mog1 in C-ter or in N-ter of MoeA likely occurred twice in Eukaryotes. Could the authors discuss the potential advantage of fusing the two proteins? Were there any cases observed in Prokaryotes?

The length of the MogA domain in the fused version of the protein in Eukaryotes is equivalent in most species (aprox. 160 residues), but the length of the connector that links it to MoeA differs. We identified a few “intact” copies of MogA in the Eukaryotes, mainly in Algae and Fungi. This further suggests that the MogA domain fused to MoeA is involved in MOCO biosynthesis in most Eukaryotes.

We reconstructed a phylogeny of all MogA proteins in the Eukaryotes, but again, the lack of signal given the few positions in the alignment (aprox. 150 positions) did not allow us to go further in the inference, so we did not include it in the original manuscript. The eukaryotic intact MogA seems to have a different origin than the eukaryotic C-ter and N-terminally fused proteins that are monophyletic. But given the limitations of the method, we have no confidence in the results and we prefer not to present them. We have now explained this in the results (lines 158-161).

Both plant Cnx1 and human gephyrin have been shown to form a network by the dimerization of the MoeA domain and the trimerization of the MogA domain. In both cases, these networks interact with the cytoskeleton, so we can hypothesize that the

fusion of the protein is necessary for the anchoring to the cytoskeleton. Regarding the two fusion events, the MogA domain in Cnx1 is fused to the domain IV of MoeA, which is the one that binds to the neuroreceptors in Gephyrin. We could hypothesize that the “relocation” of MogA to the N-terminal region of Gephyrin set the domain IV free and permitted the gain of function at the synapses. We now mention this hypothesis in the discussion (lines 425-430)

We did not identify MoeA-MogA fusions in our Prokaryotic database, suggesting that this was an Eukaryotic invention.

4) L. 264-265. Could there be some statistics to support the claim that “MoeA-PBP and tungsten transporter TupA cooccur when present”? How many genomes are supporting this for each phylum? Is this significant? There seems to be two phyla where MoeA-PBP is absent while TupA is present. Could the authors rephrase the above sentence to make it more precise? i.e., that MoeA-PBP cooccur with tungsten transporter TupA when present?

We agree in that we did not provide enough support to our claim. To confirm our results we repeated the analysis on a larger dataset, that includes all complete bacterial genomes deposited in the GenBank database of the NCBI (one representative per species). We analysed all 3503 genomes, and we identified that 437 contain TupA and 197 contain MoeA-PBP, co-occurring 170 times. In 48 cases, both proteins are encoded in the same locus. We reconstructed a phylogeny of TupA, and we indicated which copies co-occur with MoeA-PBP, either in the same locus or not. We have now added this information into the main text (lines 272-275) and provided new supplementary information (Supplementary Figure 5 and Supplementary Table 3) that supports these results, detailed by phylum. The methods have been updated accordingly (lines 510-513)

5) I am not very convinced by the PCA analysis and its signification or significance, plus the colors used (and low resolution of the panel) makes it difficult to interpret. Could the authors elaborate on the reasons of their interpretation? And indicate the contribution of the different axes? Wasn't there another, maybe more straight-forward way to compare the potential active sites of the different proteins?

When making this Figure we tried indeed many different ways to represent this information, none of them being ideal. For instance we identified which pairs of residues diverge the most in terms of distance when we compare pairs of MoeA groups (Archaea MoeA1 vs. Archaea MoeA2, Archaea MoeA1 vs. Bacteria MoeA, ...), and then mapped these pairs of residues on representative structures. However we found that it was not simple to describe the method, and that the figures were also not simple to read, as we had to pinpoint to several residues on an small active site. We found that the PCA is easy to describe in terms of methodology and easier to observe. Each point represents a linear combination of all the computed distances between the residues in the active site for an individual protein structure. We see that the distances between the residues in the active site of all eukaryotic and Archaea MoeA2 proteins group together, suggesting that they are similar, and potentially have the same enzymatic function. This is not the case for Archaea MoeA1 that, while the distances between the individual proteins are similar to those of Bacteria or Archaea MoeA1, they are grouped separately, suggesting a

potentially different function. In the case of Bacteria Glp, the distances between the individual proteins are clearly larger, indicating that the distances between the residues are not conserved, and thus the active site is not conserved suggesting a loss of enzymatic function.

We agree that the choice of color and the resolution was bad, and we now provide an improved version that is hopefully easier to read. We have also clarified the description in the figure legend to make the figure easier to interpret (lines 626-628).

Minor points

6) Figure 1. The molecules' images could be sharper. In legend, please italicize the names of all the organisms. "Domain MoeA ..." the colors in the legend and on the figure do not match. Please also define "UFB" upon 1st appearance. It seems that the tree in panel (e) is not described. Is that still the tree of MoeA/Gephyrin? Or is that a species tree? Please define Sar in the legend.

We have now corrected all these points linked to Figure 1.

7) In line 164, differential losses of MoeA are suggested in Fungi, but this does not show in Figure 1E. Is this the case for other lineages too?

In Figure 1e we show one representative species that kept MoeA. The differential losses of MoeA were evident in Fungi, so we conducted a dedicated analysis that we show in Supplementary Figure 3. For the other lineages we didn't identify evidence of differential losses in our database of the Eukaryotes so we did not investigate further each eukaryotic kingdom.

8) Please write for Sup Fig. 3 that the red dot indicates the loss. Usually, it is the presence of a trait which is symbolized.

We have now added this information.

9) L. 171 and possibly fusion of MogA in C-ter? Are we sure that Cnx1-type proteins are not also moonlighting?

We are not sure if we understood correctly the first part of the question. In the case it refers to the events observed in the Fungi, we didn't mention the fusion of MogA in the C-ter as our results suggest that the fusion happened before the appearance of the Fungi, in the Amoebozoa (or earlier).

To the best of our knowledge, the moonlighting function of Cnx-1 has not been clearly established and remains an open question. However the fact that Cnx-1 also interacts with the cytoskeleton suggests a possible analogy with gephyrin.

10) L. 307-308 are partially redundant with 302-304 concerning the archaeal homologs, the authors might want to rewrite this part a bit?

We have now removed this redundancy (lines 313-315)

11) The resolution of the structures and of the PCA panel could be improved on Figure 4a.

We apologize for the low resolution of the images, we had to compress the files for the submission because of the file size limitations. This will be fixed for the final version of the Figures.

12) L. 405. “evolutionary” instead of “evolutive”?

This has been corrected (now line 411).

13) L. 451. Could the authors explain why there is so little bacterial genomes (81) compared to archaeal (122) and eukaryotic (129) ones? Whenever bacteria are more diverse? Is the dataset used here representative of the bacterial diversity?

For both prokaryotic domains we wanted to work with a balanced dataset of five species per phylum. For this, we reused a bacterial and an archaeal sampling we published before. In that case we had 387 bacterial species, that outnumbered both the archaeal and the eukaryotic datasets. The resolution of the bacterial subtrees was significantly worse than the archaeal and eukaryotic ones, as it contained many Candidate phyla, that likely introduce noise (genomes assembled from metagenomes, usually incomplete or having bad quality), hence we decided to reduce the bacterial dataset, removing the Candidate phyla. The bacterial sampling we kept is representative of all the diversity, as it contains five species from each known phylum that had a cultured representative at the time of the analysis.

14) L. 490. *E. coli* to be in italics. Same for *S. cerevisiae* in line 478 (plus other occurrences elsewhere).

This has been corrected.

15) In Methods, lines 504-510 are mostly repeating the phylogenetic reconstruction presented in lines 478-486. Could the authors rationalize the text? The fungi tree could probably be explained in the same section than that of the other trees, especially when the “mapping” part of MoeA and MoeA homologs is basically also the same.

We have now reorganized this information in the methods section (lines 492-496 and 520-523).

REVIEWERS' COMMENTS:

We thank both reviewers for their comments and the time they have spent to improve this manuscript. The final points raised by reviewer #3 have been addressed below.

Reviewer #1 (Remarks to the Author):

The authors have adequately addressed all of my concerns from the initial review, incorporating the requested modifications into the revised manuscript. Based on these improvements, I am satisfied with the current version and recommend this work for acceptance and publication.

Reviewer #3 (Remarks to the Author):

I thank the authors for this revised version and the comprehensive responses to my questions/comments. I am fully satisfied with these and am happy to recommend this article for publication.

I only leave a few minor points/typos below, which I'm confident the authors will address.

- In Materials and Methods, line 464: please add how were selected the bacterial phyla, as you mentioned in the rebuttal that they are only representing the phyla with cultured representatives, and it is not indicated here where it should.

Done, now lines 442, 443

- Line 480: "sensitive homology searches": this is a shortcut. They are really "similarity searches", the goal being indeed by proxy to find homologues.

Done, now 458

- Line 221: "fact that both homologs are continuous" => change to "contiguous"

Done

- Line 357: "the species archaeal tree" => maybe change to "the archaeal species tree"

Done, now line 338

- Line 375: same here: "with the species bacterial tree" => change to "bacterial species tree"

Done, now line 354

- Line 494: there is a space missing.

Done, now line 471

- Some species names are still not in italics in Mat&Meth (e.g. L. 498)

We have verified all the italics again and formatted them correctly